# Make Every Example Count: On the Stability and Utility of Self-Influence for Learning from Noisy NLP Datasets

**Irina Bejan**[*] and **Artem Sokolov** and **Katja Filippova**
Google DeepMind
irinam.bejan@gmail.com, {artemsok, katjaf}@google.com

## Abstract

Increasingly larger datasets have become a standard ingredient to advancing the state-of-the-art in NLP. However, data quality might have already become the bottleneck to unlock further gains. Given the diversity and the sizes of modern datasets, standard data filtering is not straight-forward to apply, because of the multifacetedness of the harmful data and elusiveness of filtering rules that would generalize across multiple tasks. We study the fitness of task-agnostic self-influence scores of training examples for data cleaning, analyze their efficacy in capturing naturally occurring outliers, and investigate to what extent self-influence based data cleaning can improve downstream performance in machine translation, question answering and text classification, building up on recent approaches to self-influence calculation and automated curriculum learning.

## 1 Introduction

Deep learning on increasingly larger and diverse data sources brought impressive advances in natural language processing (NLP), however, data quality might be the major bottleneck to unlock further gains (Kumar et al., 2020). NLP data are usually acquired via large-scale weakly-labeled data scraping or crowd-sourcing labels from non-expert human annotators, which are both error-prone (Bowman and Dahl, 2021). At the same time, ambiguous training data are also known to hurt models' performance through overfitting or memorization in overparameterized networks (Zhang et al., 2017). Finally, not all data are equally easy to learn and overly complex instances may hinder learning as well. Below, we refer to all of those cases – label noise, out-of-distribution, ambiguous or difficult-to-learn examples – by an umbrella term *outliers*. Two key questions of our work are: How can outliers be detected and how should they be dealt with?

---

[*] The work was done while interning at Google.

## 1.1 Detecting Outliers

Defining outliers and how they may (harmfully) influence model predictions in a task-agnostic way is hard and so, until recently, mostly task-dependent heuristics have been employed (Wang et al., 2018). More principled approaches define an impact of a training instance via the concept of *influence functions* (henceforth IFs) (Cook and Weisberg, 1980), which quantify the effect on the loss on a test point $z$ when removing an individual training point $x$. For example, Koh and Liang (2017) used access to gradients and their fast products with the loss Hessian (Pearlmutter, 1994) to approximate the loss change at $z$ that would occur had $x$ been infinitesimally upweighted in the training set. IFs have been used for debugging of machine learning models (Han et al., 2020), data poisoning attacks (Koh et al., 2022) and detecting dataset errors (Schioppa et al., 2021; Kong et al., 2022). There, it has been conjectured and empirically tested that filtering highly *self-influential* ($z = x$) points, i.e., the ones that would cause a large loss delta on themselves (suggesting that they are "unsupported" by other data points and need to be memorized), does lead to improvements in synthetic and real scenarios.

To deepen this line of work, we also operationalize IFs to detect outliers with self-influence scores, and formulate our first research question being:

> RQ1: When are self-influence scores effective for detecting outliers?

The above improvements, however, contrast with the observations that IFs are sensitive to model and training hyperparameters in the general, $z \neq x$, case due to violation of the convexity assumption by IFs in deep learning: Basu et al. (2021) showed that depth and width of the network, its architecture, training regularization and the stochastic approximations inside IF have strong effects on the IF accuracy and stability (measured on retrieved influential $x$s for a fixed $z$), which are aggravated with

the network size. K and Søgaard (2021) further found that IFs are sensitive to the parameter initialization, ordering of the training data and batch size. Both papers thus doubted that IF scores of training instances would be reliable for practical purposes, and that retraining after removing or fixing the flagged instances would lead to improvements. Very recently Schioppa et al. (2023) gave a theoretical perspective on IFs instability.

Since, at a minimum, for self-influence to point at outliers in any objective and verifiable sense, they should exhibit certain stability, this leads us to the second research question of our work:

> RQ2: How stable are self-influence scores?

Unlike general influence, the self-influence stability has not been covered by previous studies.

## 1.2 Dealing with Outliers

A standard approach to reducing the harms caused by outliers is to filter them out (Khayrallah and Koehn, 2018; Peskov et al., 2019). However, coming up with a filtering rule that would generalize across multiple tasks is not straightforward. Most scalar-based (incl. self-influence) filtering schemes would prescribe setting a threshold cut-off value to delineate outliers from the rest data. This may be reasonable for datasets where the to-be-filtered data portion has no apparent signal, however, in more realistic scenarios many outliers are at least somewhat useful. Applying threshold filtering in such situation may lead to performance decrease as a portion of useful signal would be lost. For example, memorizing *some* outliers (e.g. under-represented training examples) can in fact improve accuracy (Feldman and Zhang, 2020).

Motivated by this shortcoming, in this paper we explore alternatives to filtering where it is possible to make use of outliers and in particular consider *automated curriculum learning* (AutoCL). AutoCL covers a range of algorithms, where not only the training data are presented to a neural network in a different order than random sampling, but also where this order is adapted alongside main training based on learning progress (Graves et al., 2017; Kreutzer et al., 2021). This is particularly useful when dealing with outliers, as we can learn (via the self-influence proxy) to ignore the outlying data samples and prioritize the most helpful ones, without having to choose apriori the cut-off threshold.

Thus, the final research question we address is:

> RQ3: Does AutoCL with self-influence scores bring gains compared to filtering?

## 1.3 Contributions

We study the stability of *self*-influence scores, which are task-agnostic and, if stable, would be an attractive candidate to serve as the data cleaning foundation. We further analyze the efficacy of capturing naturally occurring outliers by IFs and investigate to what extent self-influence can improve performance in NLP tasks with Transformer architectures, building up on recent improvements in IF approximation accuracy and scalability with the Arnoldi iteration based IFs (ABIF) (Schioppa et al., 2021), and AutoCL (Kreutzer et al., 2021).

In more detail, our contributions are:

- **Stability of self-influence scores.** We start by measuring how stable self-influence scores are, since this is a prerequisite for both successful data filtering and data scheduling. To this end, in §3, we study correlation and overlap of data rankings by self-influence scores across different model states, i.e., different final states the training converges to as a function of varying batch size, random seeds of data sampling, data ordering and IF calculation. We also explore the correlation between model prediction stability (defined below as *model churn*) and IF's sensitivity to architecture. We find that, unlike the general ($z \neq x$) IF scores, the self-influence ($z = x$) scores *are* stable with respect to training and model hyperparameters, and across architecture variations, but care should be exercised in transferring findings between architectures of different capacity.

- **Effectiveness of self-influence scores.** In §4, we employ a suite of different in-distribution and out-of-distribution (o.o.d. ) evaluation setups and show that filtering out highly self-influential examples is more effective for the o.o.d. setup. We hypothesize that self-influence capturing general outliers prevents learning systematic noise patterns that would otherwise artificially inflate performance in the in-distribution evaluation setup, making it harder to improve upon with filtering. Furthermore, we investigate what is captured by influence scores using both natural outliers and synthetic label noise, showing that natural data can be spread among high and low influential samples, thus the common top-X% filtering strategies can be ineffective.

- **Data filtering automation.** The fixed percentage filtering can also be costly to tune or inaccurate, while attempts to automate it using abrupt changes at the top of the ranking have a low recall rate (Lam et al., 2022). To remedy, in §5 we employ bandit AutoCL to dynamically detect, during training, the harmful or the useful training data quantiles to feed from at each step. The possible bandit actions are derived from the self-influence ranking and further split into a fixed number of discrete buckets. As a result, AutoCL adjusts on-the-fly the ratio of high or low influence examples to train on. This is more general than threshold filtering, which is a particular (static and hard-weighted) case of general (dynamic and soft-weighted) schedules.

## 2 Tasks, Datasets, Models and Methods

Throughout this study, we investigate how self-influence methods perform and generalize across multiple NLP tasks, varying the tasks' nature, size, noise levels and model architectures (Table 1).

### 2.1 Tasks and Datasets

**MT:Paracrawl.** We consider the German-English translation task from the noisy Paracrawl corpus (Bañón et al., 2020), which consists of 100M sentence pairs obtained via web-crawling. We evaluate using BLEU after a fixed number of steps on the newstest17 set from WMT17 to match the setup of Schioppa et al. (2021), who also filtered Paracrawl with ABIF.

**QA:Natural Questions.** The NQ dataset consists of real queries issued to the Google search engine, alongside Wikipedia fragments that could potentially contain an answer (Kwiatkowski et al., 2019). Each query can have a short answer (incl. empty) and a long answer, the latter requiring to predict spans from the fragments. Since we run our NQ experiments with a seq2seq model, we adopt the dataset version which only covers short answers from (Guo et al., 2022), who split the official training set of 307k samples (90% for training, 10% as the dev set) for fine-tuning, and use the official dev set for testing. From the data quality perspective, being real user queries, NQ is relatively clean but contains a high degree of natural ambiguity: about 33% of NQ annotations are debatable and 16% are wrong, meaning the Wikipedia fragment provides no evidence for the ground-truth answer (Kwiatkowski et al., 2019).

**QA:TriviaQA.** This dataset includes 110k question-answer pairs authored by trivia enthusiasts, who gathered evidence documents for answering questions drawn from Wikipedia and Bing search results (Joshi et al., 2017). This is a particularly high quality supervised task, but is still difficult to learn: the input length is on average 10 times longer than in NQ, bringing additional challenges such as complex, compositional questions that require higher cross-sentence reasoning. We evaluate both on TriviaQA and on NQ using the Exact-Match (EM) and F1 scores.

**Classification:Wikipedia Toxicity.** The dataset contains 223k human annotated comments from Wikipedia talk page comments (Wulczyn et al., 2017). While the original dataset covers a variety of toxicity subtypes, we only consider a binary classification into toxic and non-toxic comments as in (Ebert et al., 2022), and report accuracy and F1.

### 2.2 Models

We experiment with three different architectures: the standard Transformer-base on the MT task, two sizes of the state-of-the-art LongT5 architecture for long inputs on the QA tasks, and the classic BERT-base and T5 architectures on the text classification task. See §A for training details of each of those.

### 2.3 Methods

**Influence functions** are an approximation to the loss change at the test point $z$ after an infinitesimal upweighting of a training point $x$ (Koh and Liang, 2017): $I(x, z) = \langle \nabla_\Theta L(z), H^{-1} \nabla_\Theta L(x) \rangle$, where $\nabla L(x)$ is the gradient of the loss $L$ at the points $x$ or $z$, and $H = \nabla_\Theta^2 L$ is the Hessian of the model at parameters $\Theta$. For deep learning, the Hessian is impractical to compute exactly, so Koh and Liang (2017) proposed an approximate estimation procedure to calculate $I(x, z)$. Recently, Schioppa et al. (2021) proposed a more accurate and stable method that uses Arnoldi iteration to approximate the inverse Hessian in subspaces spanned by $H$'s largest eigenvectors. This enabled scaling up the computation of influence scores to hundreds of millions of training points and model parameters. We use their released code to compute $I(x, z)$ (ABIF).

TracIn (Pruthi et al., 2020) is a gradient-only alternative influence definition that relies on $C \geq 1$ checkpoints to approximate by how much $x$'s gradient changes model parameters and, in turn, the loss at $z$: $I_T(x, z) = \frac{1}{C} \sum_{c=1}^{C} \langle \nabla_{\Theta_c} L(x), \nabla_{\Theta_c} L(z) \rangle$.

| Dataset | Task | Noise | Model | Training | Architecture | Params | Train/Dev/Test |
|---|---|---|---|---|---|---|---|
| Paracrawl | MT | very high | enc-dec | from scratch | Transformer-base | 60M | 100M/3k/3k |
| Natural Questions | QA | low | enc-dec | fine-tuning | LongT5-base/large | 220M/770M | 276k/31k/7.8k |
| TriviaQA | QA | very low | enc-dec | fine-tuning | LongT5-base | 220M | 88k/11k/11k |
| Wikipedia Toxicity | text-class. | high | enc(-dec) | fine-tuning | BERT-base/T5-base | 110M/220M | 144k/16k/63k |

Table 1: Dataset and model statistics.

The question of which layers are more effective for IFs is still open, with recent work showing good results using the last or few last layers (Han et al., 2020; Barshan et al., 2020), but also using the first layers (Yeh et al., 2022). Therefore, we experiment with IF methods in three variants: *first* (the first two layers of the encoder and decoder), *last* (last two layers of both) and *all* parameters and draw a comparison between them. For the Paracrawl experiments, following Schioppa et al. (2021), *first* and *last* only include the first two encoder and the last two decoder layers; and the ABIF eigenvectors were extracted from a model trained on WMT17. More details on self-influence computation in §B.

**Self-influence and outliers.** For both influence definitions, the self-influence of a training point $x$ can be derived from them setting $z = x$. It has been conjectured that high values of self-influence indicate data outliers (Koh and Liang, 2017; Pruthi et al., 2020); intuitively, if removing $x$ deteriorates the loss value on itself, then $x$ should be different enough so that the prediction on $x$ could not be learned from the rest of the training data and had to be memorized. Grounding the influence definition in the loss magnitude covers many possible causes for being an outlier, such as mislabeling (i.e., true noise), ambiguity (i.e., multiple possible labels, depending on the information missing from the input), being out-of-distribution, or being a difficult example (for the model) for other reasons.

**Automated curriculum learning.** We use the framing of curriculum learning as a multi-armed bandit problem (Kreutzer et al., 2021), where arms represent distinct subsets of the data that are considered bandit actions and are played at each training step $t$. When an action $a^t$ is played (as mandated by the EXP3 or EXP3S algorithms (Auer et al., 2002)), a uniformly sampled batch belonging to that data subset is fed to the model and the scalar reward feedback $y^t = Y_{a^t}^t$ is received, where $Y^t$ would be an unknown reward vector of all possible actions. Through this, the bandit learns alongside the main task to minimize the regret, $R = \mathbb{E}[\sum_t y^t] - \max_a \sum_t Y_a^t$, of not having

played the best-in-hindsight arm.

To quantify the learning progress, existing metrics are looking at the loss ($\mathcal{L}$) decrease or the increase in model complexity (Graves et al., 2017). Among those, we use normalized prediction gain reward (*pgnorm*): $1 - \mathcal{L}(\theta^{t+1})/\mathcal{L}(\theta^t)$ and the cosine similarity reward between the gradients of the training and the reward batches, where the reward batches are (re)sampled from development sets, following (Kumar et al., 2019; Kreutzer et al., 2021).

## 3 Stability of Self-Influence

In this section, we evaluate the stability of self-influence scores with respect to model states, architecture and ABIF-specific hyperparameters with Spearman rank correlation and the 90th percentile overlap (i.e. overlap between top-10% examples), suggested by K and Søgaard (2021) as an alternative to global correlation since one normally cares only about highly self-influential examples.

It is important to understand whether self-influence ranking, given that it is thought to be predictive of data quality, is an inherent data property or it is mainly rooted in the architecture. In this regard, we look at the extent to which model stability and self-influence stability are interconnected, via the model *churn* metric (Cormier et al., 2016), which is the joint expected percentage of errors on a test distribution. For example, if model A is right on 9% of the examples that model B gets wrong, and B is right on the 10% of the examples that A gets wrong, the churn is 19%. While changing weight initialization does not always impact accuracy, it can result in a net zero wins and losses, referred to as unnecessary churn.

### 3.1 Dependence on model states

We investigate if the self-influence scores are sensitive to the model state, i.e., initialization of the model, data ordering or batch size. Previously, K and Søgaard (2021) showed instability of general IFs with regards to these variables, while we turn attention to *self*-influence stability, given its foundational role for data filtering.

| Layers | Method | LongT5: NQ | | Transformer: Paracrawl | |
|---|---|---|---|---|---|
| | | 90th ∩ | Spearman | 90th ∩ | Spearman |
| *first* | ABIF | 77.78 | 0.781 | 40.71 | 0.727 |
| | TracIn | 80.49 | 0.938 | 63.47 | 0.872 |
| *last* | ABIF | 87.67 | 0.933 | 51.03 | 0.726 |
| | TracIn | 86.95 | 0.949 | 64.83 | 0.901 |
| *all* | ABIF | 78.03 | 0.804 | 44.58 | 0.771 |

Table 2: Stability of self-influence estimates to changing model states (batch size, data ordering and model initialization), using ABIF and TracIn for LongT5-base on NQ and Transformer-base on Paracrawl.

**Setup.** To evaluate the sensitivity of self-influence to changes in model states, we fine-tune the same model twice: first we fix all hyperparameters and second, we vary the batch size, data ordering seed and the model initialization, keeping the rest of the hyperparameters fixed between the two runs, to look into the worst case scenario out of the ones proposed in K and Søgaard (2021). For both runs, we compute the self-influence scores for the training set and compare the two resulting rankings for all three variants of ABIF (30 eigenvectors) and TracIn. We found it too slow to evaluate TracIn when using *all* layers, so we only report results obtained with ABIF. We run this analysis for two architectures/tasks: LongT5-base on NQ and Transformer-base on Paracrawl.

**Results.** From Table 2, we see that both methods are considerably more stable to changes in the model state, than in (K and Søgaard, 2021), where the maximum 90th percentile overlap was 32.77 for IFs and Spearman correlation below 0.07. Despite that the 90th percentile overlaps for Transformer are lower, we can see the ranking correlation is still high and believe that, because Paracrawl is a very noisy dataset (>90% of it is noise, as we show below), the overlaps are less informative.

The choice of layers has a significant impact on the stability, the last layers being more stable compared to the first layers, which is consistent with previous work (Han et al., 2020; Barshan et al., 2020) where the last layer also yielded better results. We believe that these results indicate that self-influence is robust enough to be relied on in detecting training outliers.

### 3.2 Dependence on model architecture

Basu et al. (2021) found that network architecture, in particular its depth and width, impact the accuracy of IFs. Here, we investigate to what extent self-influence is sensitive to a broader set of model changes that affect model capacity and capabilities.

In order for self-influence to surface dataset error/outliers, a low degree of instability across such changes would be necessary to avoid misattribution of self-influence stability to model architecture. We compare the self-influence scores resulted from:

- LongT5-base vs. LongT5-large: we fine-tune both models with the same hyperparameters to analyze the sensitivity of self-influence to model size that increases from 220M to 770M params.

- Local vs. Transient-Global attention of LongT5: we fine-tune two LongT5-base models, each with a different attention, yet the same configuration of other hyperparameters, to analyze the sensitivity to increased capability at same model size.

**Results.** From Table 3, changing the model capacity (size or attention) has a negative effect on the stability of self-influence scores, with the size hyperparameter affecting it less. The *first* or *all* configurations make self-influence scores more stable to large capacity changes than *last* layers, which were more robust to training parameter modifications. We conclude, given the strong correlation between increase in churn and decrease in stability, that model instability is a contributor to the self-influence scores' instability. Importantly, self-influence scores appear to be particular sensitive to model's architecture or capacity, and should be used with caution across differently-powered models. This is expected, as the architecture and model capacity, unlike training hyperparameters, define the loss landscape and its dynamics under training data perturbations. Below, we hence calculate and use self-influence scores for fixed architectures to minimize the chances of running into instabilities.

### 3.3 ABIF-pertinent instability

Finally, as ABIF is a newly developed method, we inspect the effect of its hyperparameters on stability in §C and find that contributions pertaining to ABIF itself are not of concern.

## 4 Effectiveness of Self-Influence Scores

The impact of filtering highly self-influential examples on the downstream performance and the recall of synthetically perturbed training samples, have been used to measure the correctness of IFs (Guo et al., 2021; Schioppa et al., 2021), given that the ground-truth estimate via leave-one-out is unfeasible to compute even for medium-sized models. We ask whether filtering of highly self-influential

| Model A | Model B | Churn | Layer | 90th ∩ | Spearman |
|---|---|---|---|---|---|
| LongT5-base TGlobal attention $\|B\|$=**128,** $seed_{shuf/init}$=**0** | LongT5-base TGlobal attention $\|B\|$=**64,** $seed_{shuf/init}$=**43** | 8.6% | *first* *all* *last* | 77.78 78.03 87.67 | 0.781 0.804 0.933 |
| LongT5-**base** TGlobal attention $\|B\|$=128, $seed_{shuf/init}$=0 | LongT5-**large** TGlobal attention $\|B\|$=128, $seed_{shuf/init}$=0 | 12.77% | *first* *all* *last* | 68.06 67.89 42.00 | 0.630 0.621 0.432 |
| LongT5-base **TGlobal attention** $\|B\|$=128, $seed_{shuf/init}$=0 | LongT5-base **Local attention** $\|B\|$=128, $seed_{shuf/init}$=0 | 13.54% | *first* *all* *last* | 61.19 60.05 41.92 | 0.591 0.591 0.292 |

Table 3: Relation between model stability and its architecture, capacity or training hyperparameters, on the NQ dataset. Bold marks differences between models A and B. The first group of ABIF results is from Table 2.

examples is more helpful for o.o.d. evaluation (by removing true outliers) or if it can also improve performance on test sets distributed similarly to training data (and containing the same error types).

**Setup.** To evaluate the performance of filtering, we calculated the self-influence scores using all three layer settings of ABIF, sorted them to retrieve the highly self-influential examples, and experimented with different thresholds given that the ratio of (harmful) outliers in each dataset is unknown. Then we retrained on the filtered data, kept the best result across the layers choices and reported nearby percentages to illustrate the performance trend.

We consider three tasks with same distribution evaluation (on NQ, TriviaQA and Toxicity) and three o.o.d. setups (NQ, Paracrawl and Toxicity): 1) Training a Transformer-base model on Paracrawl using the same setup as above, and evaluating on the newstest17 dataset. 2) Fine-tuning the LongT5-base on NQ as before, but evaluating on the TriviaQA dataset to make it o.o.d. To align the task definitions, we only keep the normalized answer from TriviaQA's answers list, whereas usually the metrics are computed against each of the given answers and the maximum score is reported. 3) Fine-tuning on Wikipedia Toxicity, but evaluating on the o.o.d. Civil Comments (CivilC) development set of 97k comments (Borkan et al., 2019).

**Results.** From Table 4, we see that self-influence filtering using ABIF brings higher improvements for the o.o.d. evaluation setup: up to +9 BLEU points on Paracrawl-newstest17 and +3 F1 points on NQ-TriviaQA setup, with a negligible improvement in the Toxicity-CivilC case, which shows that training on cleaner datasets improves performance on an o.o.d. test set. In the in-distribution setup, TriviaQA and NQ trained on full data always outperform filtering, which is not surprising given both are high-quality datasets, but also brings very small

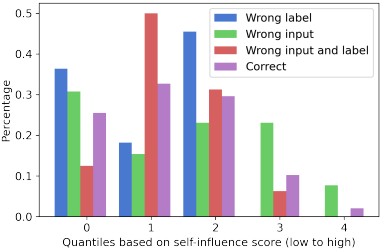

Figure 1: Distribution of the correct and incorrect examples annotated by expert annotators in 5 equally-sized bins (quantiles) computed based on the self-influence ranking (*last* layers), and ordered from low (0) to high influence (4). The annotations include wrong input (context), wrong label, or both wrong.

improvements on Toxicity, which is expected to be very noisy. This shows that the common heuristic of filtering highly self-influential examples might not be the best fit for the in-distribution tasks and we develop further why.

### 4.1 Noise captured by high self-influence scores

Here we study how a naturally occurring noise, as annotated by human experts, is partitioned by the self-influence ranking. Additionally, we compare to synthetic noise, tested previously to be accurately retrieved by high self-influence in (Schioppa et al., 2021), to see if significant differences occur.

**Setup.** We use the 5-way annotations by human experts on a set of 205 uniformly sampled NQ examples and released with the original paper (Kwiatkowski et al., 2019). The examples were annotated as correct, debatable and wrong, but we treat the debatable entries to be in the correct bucket as there is no strong evidence that suggests they are wrong. We compute the self-influence scores on the *first* and the *last* layers and analyze how the natural noise is being ranked.

For comparison, we looked at the ability to

| | Train | Eval | % Used | EM | F1 | BLEU |
|---|---|---|---|---|---|---|
| seq2seq | NQ | NQ | **100%** | **59.05** | **63.97** | - |
| | | | 90% | 58.49 | 63.06 | - |
| | TriviaQA | TriviaQA | **100%** | **78.08** | **80.17** | - |
| | | | 98% | 77.49 | 79.72 | - |
| | | | 90% | 74.64 | 76.96 | - |
| | NQ | TriviaQA | 100% | 16.06 | 20.55 | - |
| | | | **95%** | **18.96** | **23.52** | - |
| | | | 90% | 17.52 | 21.85 | - |
| | Paracrawl | newstest17 | 100% | - | - | 21.36 |
| | | | **10%** | - | - | **30.45** |

| | Train | Eval | % Used | Acc | F1 |
|---|---|---|---|---|---|
| classification | Toxicity | Toxicity | 100% | 92.61 | 95.80 |
| | | | **95%** | **93.15** | **96.13** |
| | | | 90% | 92.86 | 95.97 |
| | Toxicity | CivilC | 100% | 95.28 | 97.51 |
| | | | **90%** | **95.42** | **97.59** |

Table 4: Performance of percentile filtering of highly self-influential examples as per ABIF self-influence on in- and out-of-distribution test sets, for seq2seq and classification tasks. We report the maximum over *all*, *first*, and *last* settings. For the seq2seq tasks, the metrics that don't apply are marked with dashes.

retrieve synthetic noise via the self-influential examples and compute the recall in the top-10%/20%/30% of self-influence scores. We alter the original data by uniformly sampling 10% of the dataset for which we shuffle the labels, ensuring all labels were changed from its initial value. This is important because a significant amount of questions have no answer (they are not answerable), because they are natural search queries.

**Results.** The synthetic noise retrieval confirms previous findings with a high recall, as 29% of the synthetic noise lands in the top-20% ranks and 94% in the top-30%, when using *last* layers (vs. 84% in top-30% for *first* layers). We hypothesize that the synthetic noise is not predominantly in top-10% because other forms of outliers are already present in the dataset that are more harmful to the model.

The behaviour of natural noise is considerably different. It barely comes up among highly self-influential buckets as we can see in Figure 1, but is distributed predominantly among the lowest and mid-influential buckets. Examples annotated as having wrong labels are absent from the top-20%. Additionally, we see that input noise does not affect the model as much as label noise, given that examples with wrong input are almost evenly distributed in the lower 80% of the data. These results suggest that manual tuning of the filtering threshold of self-influence rankings may not find a percentile range which, if removed, improves performance.

## 5 Automated Filtering of Outliers

As we showed above, using self-influence to filter noisy training data has the drawback of the difficulty of choosing a right cut-off threshold. Yet, trial-and-error search for a fixed threshold based on the downstream performance remains popular, which is costly and does not generalize across

datasets. To rectify, Lam et al. (2022) attempted clustering and identifying abrupt changes in the magnitude of the sorted self-influence scores, but this resulted in a low recall rate. Hence, we move to automated curriculum learning to dynamically identify the outlying parts of data based on the buckets of self-influence scores without committing to completely remove any of them.

First, we validate how different the quality signal is from each individual self-influence bucket on NQ. We found (Figure 4 in §F) that EM of the highest self-influence bucket is much lower (although not zero) compared to the rest of buckets, which are in turn difficult to separate, possibly, because they contain data of mixed usefulness. A positive aspect of AutoCL is that, due to exploration, the model regularly visits all buckets, and may dynamically up- or down-weight buckets depending on the model's current needs, overcoming the rigidness of a fixed threshold filtering.

**Setup.** We verify the feasibility of AutoCL with two definitions of the self-influence, given by ABIF and by TracIn, and check that the findings are consistent across three datasets (NQ, Paracrawl and Toxicity on T5). Bandit actions are mapped to equal-sized data buckets corresponding to a predefined number of percentile ranges of self-influence scores. We first explore 10 buckets, which should allow the bandit to at least match the performance of filtering, and then consider more granular setups (20 and 25 buckets) which would be infeasible to manually test against filtering. We expect the bandit not to use much of the high self-influential buckets, nor the lowest bucket, which prior work found to be less useful because of its simplicity (Feldman and Zhang, 2020; Schioppa et al., 2021). Following (Schioppa et al., 2021), we report BLEU for Paracrawl at 10k and 200k steps. As baselines

| | Config | Method | BLEU@10k | BLEU@200k |
|---|---|---|---|---|
| **Paracrawl** | Baseline | | 13.75 | 21.36 |
| | Filtered 90% | ABIF | 26.8 | 30.45 |
| | Filtered 90% | TracIn(1) | 22.10 | 27.87 |
| | AutoCL - 5 bins | ABIF | 21.45 | 27.48 |
| | AutoCL - 10 bins | ABIF | 25.50 | 30.45 |
| | AutoCL - 25 bins | ABIF | 24.13 | **31.38** |
| | AutoCL - 25 bins | TracIn(1) | 18.60 | 29.33 |

| | Config | Method | EM | F1 |
|---|---|---|---|---|
| **Natural Questions** | Baseline | | 59.05 | 63.97 |
| | Filtered 10% | ABIF | 58.49 | 63.06 |
| | Filtered 10% | TracIn(3) | 58.09 | 62.75 |
| | AutoCL - 10 bins | ABIF | **59.72** | **64.33** |
| | AutoCL - 25 bins | ABIF | 59.20 | 64.01 |
| | AutoCL - 25 bins | TracIn(3) | 59.59 | 64.32 |

| | Config | Method | Acc | F1 |
|---|---|---|---|---|
| **Toxicity** | Baseline | | 91.73 | 67.37 |
| | Filtered 5% | ABIF | 91.80 | 67.22 |
| | AutoCL - 10 bins | ABIF | **93.61** | **70.56** |
| | AutoCL - 25 bins | ABIF | 92.09 | 67.58 |

Table 5: Performance of threshold filtering and AutoCL, on top of self-influence scores by ABIF or TracIn (number of checkpoints $C$ in brackets).

we use the filtering methods from §4. See §G for AutoCL hyperparameters of reported results.

**Results.** From Table 5, we see that AutoCL strongly outperforms filtering methods when given enough bucket granularity, for very noisy Paracrawl, noisy Toxicity and low noise NQ, and regardless of the task and IF definition. We improve over filtering on Paracrawl by +1 BLEU point, on Toxicity by +3.2 F1 points, and on NQ by +1.2 F1 points. In addition, on Paracrawl at 10k steps filtering with a good threshold outperforms AutoCL which requires time to learn which are the useful buckets. Given that ABIF and TracIn showed similar results for NQ and Paracrawl, we only look at ABIF for Toxicity.

We check if the multi-armed bandit's learned policy (probabilities of choosing a data bucket) is interpretable in Figure 2. In general, there is no incentive for the policy to be interpretable as it targets loss improvements only and may undertake redundant switches between neighboring buckets in setups with high bucket granularity with the same end performance. That said, for Paracrawl, the model quickly learns to drop the top-92% of the data as ranked by self-influence, which almost matches our best filtering threshold of 90%, instead training on ⅔ of time on the bottom-4%/8% and only ⅓ of time from the top-4% which corresponds to the lowest influence and is known to be less useful to the model (Schioppa et al., 2021). For NQ, there is more of bucket switching, and the

model initially uses the mid-influence buckets (top-50%/80%), followed by the high-influence outlier buckets (top-80%) which are quickly dropped and continues alternating between the lowest buckets (in bottom-20%). For Toxicity, the policy is not interpretable (though the high influence bucket is heavily used at all stages) but still brings more gains compared to the filtering. As TriviaQA is a very clean human-curated dataset, filtering does not improve over baseline, and AutoCL brings only nominal insignificant gains (see §D).

Finally, one might wonder if AutoCL improvements are due to self-influence or due to the increased data scheduling flexibility, i.e., if a simpler example difficulty signal would suffice to attain similar gains. In §E, we run AutoCL on NQ/TriviaQA buckets defined via difficulty signals based on domain knowledge (context length, word rarity, context-question lexical overlap and question type) and find that self-influence is indeed crucial for the observed gains.

## 6 Related Work

**Dataset error detection.** Deep learning models' performance can be sensitive to noisy datasets (Kumar et al., 2020; Zhang et al., 2017; Khayrallah and Koehn, 2018), and various instance-based scoring methods have been proposed, alongside with sorting and thresholding to filter out noisy training example, including bilingual cross-entropy difference to rank parallel corpora (Axelrod et al., 2011), "forgetting events", where an individual training example transitions from being classified correctly to incorrectly over the course of learning (Toneva et al., 2019), or area-under-margin (Pleiss et al., 2020), computed as the difference between the logits value in classification tasks. IFs were used to detect dataset errors (Koh and Liang, 2017; Schioppa et al., 2021), by looking at self-influence – how much a training point influences its own loss. However, these methods led to various degrees of success: in the case of IFs, Koh and Liang (2017) and Guo et al. (2021) improve performance by directly fixing/augmenting the mislabeled highly self-influential training examples, but filtering lowest or highest influential examples out did not outperform training on full data in other studies (Guo et al., 2021; Kocijan and Bowman, 2020). At the same time, it brought consistent performance gains on an o.o.d. task for MT (Schioppa et al., 2021), raising the question whether filtering helps for improving

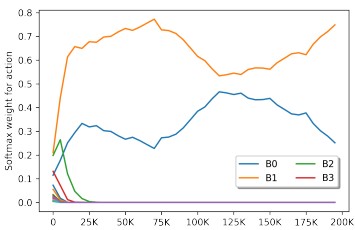 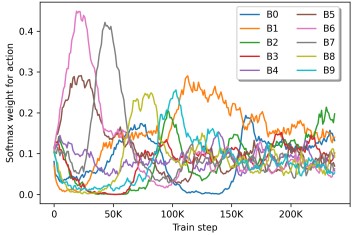 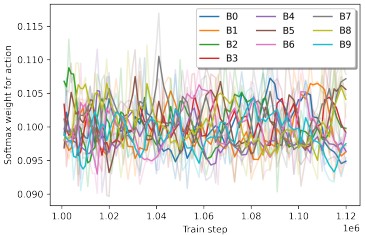

(a) Paracrawl (25 bins); very high noise.          (b) NQ (10 bins); low noise data.          (c) Toxicity (10 bins); high noise data.

Figure 2: Learned AutoCL policies, showing the probabilities attributed to each bucket (where B0 is the lowest influence bin). The policy extracted from Paracrawl (a) used 25 bins (with only B0-B3 in the legend), and the ones from NQ (b) and Toxicity (c) used 10 bins.

performance on in-distribution test sets or is more effective for o.o.d. generalization.

**Harmful vs. useful outliers.** A shortcoming of filtering/data selection is that outliers are not always harmful. Prior work has shown that training with noise, or with injected artificial noise, can increase the stability of the models vis-à-vis noisy data (Vaibhav et al., 2019; Belinkov and Bisk, 2018; Heigold et al., 2018), which can be caused by differences in noise types and their interaction with the target task. Al Sharou et al. (2021) noted that only "harmful noise" that affects the performance of the system or does not carry the intended meaning should be removed, while "useful noise" should be kept or even added to the training data if it naturally occurs at test time (Rolnick et al., 2017). Moreover, the performance of the noise detection methods were evaluated, due to a lack of suitable datasets, on synthetic noise, but Jiang et al. (2020) found synthetic noise to affect much more DNN's capability to generalize than real noise. We are particularly interested in whether that holds true for IFs and analyze what kinds of outliers are captured by highly self-influential examples.

**Dynamic data schedules.** Following this limitation of filtering methods, different training schedules that account for the training dynamics have been developed inspired by Bengio et al. (2009) and van der Wees et al. (2017). Swayamdipta et al. (2020) proposed a "easy-to-hard" training schedules based on the mean (confidence) and standard deviation (variability) of the gold label probabilities over the training epochs, where *hard* points to samples with low confidence and low variability. Wang et al. (2018) proposes using a small amount of trusted data to help the model measure

noise and do online data selection to train on gradually noise-reduced data batches, resembling active learning. Similarly, Nguyen et al. (2020) uses a self-ensemble for label filtering during training, by gradually allowing supervision from clean labels and stopping learning on the filtered noisy labels. Jiang et al. (2020) develops a method that use curriculum learning and vicinal risk minimization to handle both real and synthetic noise. We note that curriculum-based methods have been more effective than filtering, but also have inherent complications, such as defining "easy" and "hard" or designing an effective training schedule following these definitions. To overcome this limitation, we used automated curriculum learning (Graves et al., 2017; Kreutzer et al., 2021) that employ a multi-armed bandit to learn the most effective training schedule.

## 7 Conclusion

We proposed a general method for improving model performance in the presence of noisy training data based on self-influence and bandit curriculum learning, and without relying on threshold filtering. We showed that Arnoldi iteration based self-influence scores of training instances are stable with respect to varying training hyperparameters such as batch size, random seeds, and the iteration's hyperparameters, and thus can be a reliable foundation for data cleaning methods. We further demonstrated that not all data outliers (as per human annotation) receive similarly-valued self-influence scores, what necessitates generalizing threshold filtering to a dynamic data reweighing. Finally, we showed that dynamically reweighing based on multi-armed bandits, which pick self-influence bins to sample batches from, outperforms threshold filtering on noisy and clean datasets.

## 8  Limitations

A potential limitation of our approach is the requirement to precompute self-influence scores on a pretrained model and to retrain again using these scores. While recomputing the self-influence scores during training may make them reflect the influence w.r.t. the current state of the evolving model better, doing this efficiently would be a non-trivial engineering problem. Additional unexplored factors are the choice of the training data for the scores-producing model (i.e. trained on clean data, as in (Schioppa et al., 2021) and in our Paracrawl experiments, or on the to-be-cleaned data as in our QA and toxicity experiments) and the trade-offs of bandit rewards which influence the overhead of bandit learning alongside the main training (from negligible for the cosine similarity reward to an additional forward pass for the *pgnorm* reward). Finally, we left exploration of the impact of natural noise on performance, task-specific rewards and exploiting other signals for filtering to future work.

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

## A    Model training details

We experiment with three different architectures: the standard Transformer-base on the MT task, two sizes of the state-of-the-art LongT5 architecture for long inputs on the QA tasks, and the classic BERT-base and T5 architectures for text classification.

**Transformer.**    For the MT task, we implement the standard, 6-layer, Transformer-base architecture (Vaswani et al., 2017) trained for 200k steps with a batch size of 128 using T5X (Roberts et al., 2022) on 16-core TPUv2 and fixed input length of 128 on Paracrawl dataset. For training, we use Adafactor with a learning rate of 2.0 and the rsqrt decay (Vaswani et al., 2017) of 0.8, a dropout rate of 0.1 and a 32k SentencePiece model.

**LongT5.**    For our QA tasks, we use the state-of-the-art LongT5-base and LongT5-large architecture (Guo et al., 2022)[1] implemented in T5X, using a 32k SentencePiece model, a batch size of 128 and AdaFactor as in the original work. We fine-tune the models on NQ for 261k steps on a 16-core TPUv2 to convergence, with a learning rate of 0.001, dropout rate of 0.1 and a fixed input length of 4,096. We employ the same setup for TriviaQA, except we double the input length, to account for TriviaQA's much longer contexts than in NQ.

**BERT.**    For the text toxicity classification, we experiment with BERT-base (Devlin et al., 2019), and fine-tune it for 35k steps with early stopping, batch size of 32, learning rate of $10^{-6}$, weight decay of $5 \cdot 10^{-6}$, input length of 128 and dropout of 0.5.

**T5.**    To use our T5X AutoCL implementation, we reframed the toxicity classification task as a seq2seq task in T5 (Raffel et al., 2019) by predicting two tokens: *toxic* and *safe*, treating other output tokens as misclassifications, and report F1 for the toxic class. The T5-base model is trained for 120k steps, with input length of 128, batch size of 64, dropout rate of 0.1, using Adafactor, with a learning rate of 0.01 and decay rate of 0.1.

## B    Self-influence calculation

**ABIF self-influence.**    To trade-off between memory/speed for accuracy, ABIF introduces several hyperparameters that may impact its accuracy, including the choice of layer's gradients to use, the

---

[1] github.com/google-research/longt5.

---

| Layers | 90th $\cap$ | Spearman |
|---|---|---|
| number of eigenvectors | | |
| *first* | 96.79 | 0.99 |
| *last* | 96.74 | 0.99 |
| *all* | 93.90 | 0.99 |
| initialization | | |
| *first* | 96.21 | 0.99 |
| *last* | 96.92 | 0.99 |
| *all* | 97.08 | 0.99 |

Table 6: Stability of self-influence estimates with respect to ABIF hyperparameters using LongT5 on NQ.

number of top eigenvalues to use, and the number of Arnoldi iterations. We use 30 eigenvalues and 60 iterations for computing the self-influence scores, and compare it, resp., to 100 and 200 in an ablation.

**TracIn self-influence.**    We employ TracIn as a baseline for NQ self-influence scoring, and use a fixed projection size of 1,024 and three checkpoints: from the beginning of the training (60k steps), middle of training (140k) and the final checkpoint (260k). We could not scale TracIn to 100M Paracrawl examples, so we use its variant by (Schioppa et al., 2021), who reduce gradient dimensionality to 30 using Gaussian matrices for the last checkpoint at 100k steps.

## C    ABIF-pertinent instability

Finally, given that ABIF is a newly developed method, we inspect if different choices of hyperparameters for ABIF can contribute to the instability, including layers choice, number of top eigenvectors (30 vs. 100) and the initialization seed, by recomputing the self-influence scores with various configurations using the fixed hyperparameters variant of the LongT5-base fine-tuned earlier.

**Results.**    From Table 6 we can see that contributions that pertain to ABIF itself are not of concern since there are largely unaffected by the different choice of its parameters. The 90th percentile overlap is lower, despite the almost perfect correlation, because the overlap is sensitive to insignificant movements in the vicinity of the cut-off value.

## D    AutoCL on clean dataset: TriviaQA

In Table 4, we saw that filtering for the in-distribution task for TriviaQA proves unsuccessful, attributable to TriviaQA being a very clean human-curated dataset. Regardless, we attempted

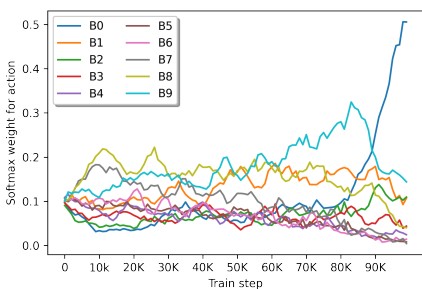

Figure 3: Policy learned by the model during AutoCL, showing the soft-max weights attributed to each bin (where B0 is the lowest influence bucket). This policy leads to same downstream quality as the full data and training with uniform sampling.

to run the AutoCL on it and found that the method reaches the same quality as the model trained on the full data (Table 7), despite not using uniformly random samples. Figure 3 shows the learned policy for 10 self-influence buckets, where we see that the model uses all buckets, initially with more emphasis on *harder* buckets (top 30% of highly influential examples) and follows up with using more the top 10% and most of the lowest influence bucket. Commencing with highly influential examples is reasonable because, given the lack of noise, they are presumably difficult examples with high impact on accuracy, which need to be memorized.

| Config | Method | EM | F1 |
|---|---|---|---|
| Baseline | | 78.08 | 80.17 |
| Filtered 10% | ABIF | 77.49 | 79.72 |
| AutoCL - 10 bins | ABIF | **78.13** | **80.29** |

Table 7: Filtering performance and AutoCL on TriviaQA (very clean dataset) using ABIF self-influence scores.

# E   Other difficulty signals for AutoCL

In §5, AutoCL delivered important gains on NQ (+1.2 F1) driven by self-influence ranking as a signal. However, one might wonder if the improvements could be attainable by AutoCL with much simpler example difficulty proxies for splitting the data into buckets.

**Setup.**   We propose a suite of heuristics, inspired by prior work, to split the training data in a fixed number of buckets, as we did with the self-influence ranking, and explore a variety of hyperparameters configurations, using the LongT5-base model on NQ and TriviaQA. We train the same

LongT5-base as in earlier sections, with the exception of training on NQ for 200k steps. The signals we employ are:

- **Context length:** (Platanios et al., 2019) proposed using sentence length as a proxy for translation difficulty. We consider the length of the context because longer context would require more cross-sentence reasoning to find the answers.

- **Word rarity:** Another layer of difficulty can be added by very specific domain-questions (thus words that rarely appear in training). According to Platanios et al. (2019), low frequency words make it difficult to learn a robust representation of the words, but also make the gradients of the rare word embeddings to have high variance. Therefore, Platanios et al. (2019) proposed using the likelihood of the unigram probabilities to aggregate the word frequencies in a difficulty heuristic. Given a corpus of sentences, $s_i{}_{i=1}^M$, the relative word frequencies are defined as:

$$\hat{p}(w_j) \triangleq \frac{1}{N_{\text{total}}} \sum_{i=1}^{M} \sum_{k=1}^{N_i} \mathbb{1}_{w_k^i = w_j}, \qquad (1)$$

where $j$ indexes unique words in the corpus. These are aggregated using the logarithm of word probabilities to prevent numerical errors:

$$d_{\text{rarity}}(s_i) \triangleq - \sum_{k=1}^{N_i} \log \hat{p}(w_k^i) \qquad (2)$$

- **Context-question lexical overlap:** Sugawara et al. (2018) showed that questions with low lexical overlap with the context tend to require reasoning skills compared to superficial word matching, and showed the models have worse performance on the subset where the answer was not present in the most similar sentence to the question. We propose as a metric the overlap of tokens between context and question after removing the stop words and normalizing by the number of remaining tokens:

$$d(q_{\text{no-sw}}, c) \triangleq \frac{\mid q_{\text{no-sw}} \cap c \mid}{\mid q_{\text{no-sw}} \mid}, \qquad (3)$$

where $q_{\text{no-sw}}$ denotes question from which stop words have been removed.

- **Question type:** Gardner et al. (2019) argued that question answering encompasses a broad range

of conceptual tasks which are posed as questions (i.e. fitting the question format), but do not resemble a cohesive task. Rogers et al. (2023) grouped these tasks into three categories, with an increasing degree of difficulty based on easiness to replace the questions in a dataset with content-free identifiers: classification (*What is the sentiment of [X]?*), slot-filling/template with no meaningful question understanding (*When was [Person] born?*) and open-ended questions.

We capitalize on that distinction for automation purposes and adapt it to the NQ-specific query types. We consider the top-N type of questions with a minimum representation of 5% of the data given by the first token of each sentence (*'who', 'what', 'when', 'where', 'how', 'which', 'will'*) and place remaining training examples in the category *other*, which should be highly diverse and contain many under-represented examples.

| | Signal | Algorithm | # buckets | Reward | EM | F1 |
|---|---|---|---|---|---|---|
| Natural Questions | Baseline | | | | **58.10** | 62.65 |
| | Length | EXP3 | 5 | pgnorm | 57.79 | **62.88** |
| | Length | EXP3S | 10 | cosine | 58.03 | 62.69 |
| | Word rarity | EXP3 | 5 | pgnorm | 57.82 | 62.52 |
| | Word rarity | EXP3S | 5 | cosine | 57.56 | 62.65 |
| | Lexical overlap | EXP3 | 5 | pgnorm | 58.10 | 62.82 |
| | Question type | EXP3 | 8 | pgnorm | 58.02 | 62.66 |
| TriviaQA | Baseline | | | | **78.08** | **80.17** |
| | Length | EXP3 | 5 | cosine | 76.14 | 78.30 |
| | Word rarity | EXP3 | 5 | cosine | 77.92 | 80.00 |
| | Lexical overlap | EXP3 | 5 | pgnorm | 77.37 | 80.01 |

Table 8: Results on LongT5-base using AutoCL on a variety of feature splits on NQ and TriviaQA.

**Results.** In Table 8, regardless of the signal chosen for the split, none of the methods strongly improves on the base model performance. This shows that self-influence scores are not only a generalizable metric to be used in junction with curriculum learning methods, but are also more informative than trivial, NLP-specific, difficulty metrics.

## F Training on different buckets in isolation

Here, we validate how different the quality signal is from each individual self-influence bucket. In Figure 4, we see that the performance of the highest self-influence bucket is much lower (although not zero) compared to the rest of buckets, which are in turn more difficult to separate, possibly, because they contain data of different grades of usefulness. A positive aspect of AutoCL is that, due to exploration, the model will regularly visit all buckets,

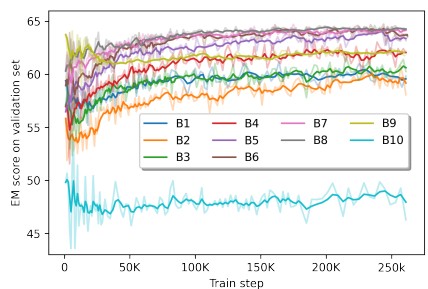

Figure 4: LongT5-base performance trained individually on each self-influence bin, from B1 (bottom-10%) to B10 (top-10%), of the NQ dataset.

and may dynamically up- or down-weight buckets depending on the model's current needs, overcoming the rigidness of a fixed threshold filtering.

## G Hyperparameters of AutoCL runs

Here, we report the hyperparameters of the bandit-based AutoCL in Table 9. All experiments used the fixed bandit learning rate of 0.001 and the exploration rate of 0.01. We tuned between two bandit algorithms (EXP3 and EXP3S) and two rewards function (*pgnorm* vs. *cosine* similarity between gradients of the train and the reward batch).

We found that for Paracrawl (very noisy dataset), the EXP3's design goal of minimizing the regret over the whole history by finding a single best arm (Auer et al., 2002) biases the bandit to commit fully to playing the current best arms (lowest influence buckets in Figure 2) and it does not reconsider noisy buckets which may decrease performance. However, for datasets where the highly influential buckets prove valuable and generally have a lower level of noise, we saw that EXP3's piecewise stationary behavior makes better use of all data and brings more consistent gains.

| | Run | Method | Hyperparameters | |
|---|---|---|---|---|
| | | | Algorithm | Reward |
| Paracrawl | AutoCL - 5 bins | ABIF | EXP3 | cosine |
| | AutoCL - 10 bins | ABIF | EXP3 | cosine |
| | AutoCL - 25 bins | ABIF | EXP3 | pgnorm |
| | AutoCL - 25 bins | TracIn(1) | EXP3 | pgnorm |
| NQ | AutoCL - 10 bins | ABIF | EXP3S | pgnorm |
| | AutoCL - 25 bins | ABIF | EXP3S | pgnorm |
| | AutoCL - 25 bins | TracIn(3) | EXP3S | pgnorm |
| Tox. | AutoCL - 10 bins | ABIF | EXP3S | cosine |
| | AutoCL - 25 bins | ABIF | EXP3S | cosine |

Table 9: Hyperparameters for the reported AutoCL results in Table 5 in §5 found with grid search. The number of checkpoints, $C$, for TracIn is in brackets.