# OpenReview forum: "Make Every Example Count: On the Stability and Utility of Self-Influence for Learning from Noisy NLP Datasets"
_EMNLP/2023/Conference — EMNLP 2023 Main_

### Official Review · Reviewer_2t6t · 2023-08-05

**Soundness:** 4

**Excitement:**

4: Strong: This paper deepens the understanding of some phenomenon or lowers the barriers to an existing research direction.

**Missing References:**

None that I'm aware of.

**Paper Topic And Main Contributions:**

This paper investigates the impact of influence-functions in the context of identifying self-influential examples (i.e. examples that are difficult to predict from other examples and are thus hypothesised to be memorised). The paper investigates properties of such self-influential examples, suggesting that they are properties of data rather than dependant on model architecture and initialisation. Finally, the paper shows how knowledge about such examples can be used to improve training and out-of-distribution generalisation of models.

**Questions For The Authors:**

Question 1: Are you going to release the experimental setup to the public?

**Reasons To Accept:**

The contribution seems novel, the experiments are (for the most part) well executed and motivated, the research questions are clear and the results support the findings.

I appreciate the fact that some of the less well-known techniques, metrics, etc by examples.

**Reasons To Reject:**

I have two concerns with this paper.

Firstly, I don't think the results regarding the stability of influence functions would withstand a statistical test - only two results are being compared here (although for different settings). Perhaps the chosen different initialisation/model happened to produce a similar result by chance. I believe for more robust conclusions, the sample set of observations should be more than a pair. I appreciate that these experiments are computationally expensive, perhaps a bigger sample could be investigated for a subset of the investigated settings (e.g. 'first' only in Table 2)

Secondly, I find the paper's content arrangement rather odd, see my comments in the presentation section. But this is only a minor concern and could be addressed by an additional page of content, should the paper be accepted.

**Reproducibility:**

3: Could reproduce the results with some difficulty. The settings of parameters are underspecified or subjectively determined; the training/evaluation data are not widely available.

**Reviewer Confidence:**

4: Quite sure. I tried to check the important points carefully. It's unlikely, though conceivable, that I missed something that should affect my ratings.

**Typos Grammar Style And Presentation Improvements:**

I find it odd to move the related work section into the appendix, while spending a whole page on Sections 2 and 3 and 4.1, which are essentially background. Sections 3 and 4.1 could definitely be moved to the appendix, since they're already summarised in Table 1.

Furthermore, the introduction set's the scene rather awkwardly. I believe the concept of self-influence should be (at least intuitively) introduced earlier and followed up by the research questions this paper is addressing. I am not sure why the paragraphs in lines 64 to 96 are so prominent.

---

> ### Author Rebuttal · Authors · 2023-08-29
>
> We appreciate your high evaluation of the work and thank you for the constructive suggestions!
>
> 1. Statistical test: Yes, experiments require costly full retraining plus rerunning the IF analysis on the full training data. On top of that, the stability metrics – 90th overlap and Spearman correlation – are easily applicable only to pairs of runs. Time permitting, we will generate more pairs and report the variance of the results.
> 2. Content arrangement: We will try to rearrange the introduction, explain IF earlier, and move the related work to the main body of the paper if it’s accepted. Thank you!
> 3. Open-sourcing: Yes, we are looking into it, although cleaning up and figuring out licensing may delay it past the conference date.

---

### Official Review · Reviewer_SJBj · 2023-08-05

**Soundness:** 3

**Excitement:**

3: Ambivalent: It has merits (e.g., it reports state-of-the-art results, the idea is nice), but there are key weaknesses (e.g., it describes incremental work), and it can significantly benefit from another round of revision. However, I won't object to accepting it if my co-reviewers champion it.

**Missing References:**

Second-Order Stochastic Optimization for Machine Learning in Linear Time, Agarwal et al., 2017.

**Paper Topic And Main Contributions:**

The paper studied whether self-influence scores based on Arnoldi-based Inﬂuence Function (ABIF) can be used to filter out low-quality data (outliers, e.g. noises) so as to improve the test performance. Specifically, the authors explored 1) the stability of computed self-influence scores against different a) model states (i.e. initialization, data ordering, and batch size) and b) model size and architecture, 2) whether self-influence scores can be used to select noises, and 3) whether one can use self-influence scores to divide datasets into different subsets, and then use Automated Curriculum Learning (AutoCL) to improve training performance. The results show that 1) self-influence scores are more robust against model states than sizes and architectures, 2) self-influence scores are better at capturing synthetic data noises than natural noises, and 3) AutoCL help learning better than filtering, based on self-influence scores.

**Questions For The Authors:**

* Do you think high self-influence data instances always harmful for models? I am thinking of large language models --- one would need LLMs to memorize certain facts to give correct answers; in this case, such memorization/high self-influence can become an advantage?
* Do you have any hypothesis why it is this case, that first/all are less sensitive to size/attention, but more sensitive to model states?
* What are the fundamental difference between synthetic and natural noises that leads to the successful filtering only on synthetic data?

**Reasons To Accept:**

* It is an interesting idea to use self-influence score to filter data.
* Understanding instability is important for such filtering to put into practice.
* Using AutoCL instead of filtering is intriging, and the performance shown is encouraging.

**Reasons To Reject:**

* The experiment setups are not well-explained, and thus not reproducible/convincing
    * Lines 360-362 should be more specific: what are the specific choices of hyper-parameters (for reproducibility)? How would that change performance (to better ground the stability)?
    * In table 4, how did the authors choose different percentage of data to filter out?


* The writing of the background/methods is hard to follow, important points include:
    * Is the IF approximation method proposed by Koh and Liang (2017)? I think the HVP method was proposed quite early.
    * Without reading Section 7, it is hard to understand what are the different subsets of data in AutoCL
    * Math notation system in AutoCL is used without introduction: what are y/Y? Also, in pgnorm (I think \mathcal L is loss?), on which data the loss should be computed; and what are the gradient and reward batch?
* The results of synthetic noises vs. natural noises detection confuse me: on the one hand, data filtering based on self-influence improved OOD performance (Table 4); however, on the other hand, such filtering failed to filter out natural noises (Figure 1) — what are the implications then? Reading the full results, I feel some pieces of evidence are missing here: self-influence scores can help filter out noisy data and help with AutoCL, however, it cannot really detect natural noises, which is a bit puzzling.

**Reproducibility:**

2: Would be hard pressed to reproduce the results. The contribution depends on data that are simply not available outside the author's institution or consortium; not enough details are provided.

**Reviewer Confidence:**

4: Quite sure. I tried to check the important points carefully. It's unlikely, though conceivable, that I missed something that should affect my ratings.

**Typos Grammar Style And Presentation Improvements:**

A bit strange to mention a section (5.3) and put everything in the appendix. It would be better give some more information in the main text for the purpose of being self-contained.

---

> ### Author Rebuttal · Authors · 2023-08-29
>
> Thank you for your thoughtful comments and questions.
>
> 1. Experiments:
>     * Thank you for pointing out the confusing line, we will reword this part to be unambiguous that in this experiment we modified only the three mentioned hyperparameters (batch size, data ordering seed and initialization seed) and the rest of hyperparameters remain the same between the two runs. The specific choice of the three hyperparameters was driven by prior work and them being the most susceptible to modification during retraining after a round of data cleaning.
>     * In all our tasks, the filtering performance decreased monotonically outside the optimal threshold thus we approached the search for it similarly to binary search, with a granularity of minimum 5%, and reported the nearby percentages to illustrate the performance trend and its decrease speed.
>
> 2. Background:
>     * You are right, the HVP method was proposed by [Pearlmutter (1984)](https://core.ac.uk/download/pdf/297017473.pdf). [Koh & Liang (2017)](https://arxiv.org/abs/1703.04730) proposed to use it and LISSA for their IF approximation. We will add the Pearlmutter citation to the text.
>     * Thanks for pointing this out - we will add more examples/explanations earlier.
>     * Thanks again - we will explain missing notation and terms, which fell prey to our draft shortening. The gradient and rewards batches are two regular SGD mini-batches used to calculate, respectively, the network update gradient and bandit rewards. In principle, they could be the same but ([Kreutzer et al., 2021](https://arxiv.org/abs/2110.06997)) found that drawing the reward batch from a held out set works better.
>
> 3. Results: Please note that the natural noise experiment is done on NQ which is a low noise, human curated, dataset. There, indeed, self-influence fails to cleanly capture the natural (~human) noise and improvements from threshold filtering are absent (Table 4, first line). AutoCL though, because it’s able to use all the data, managed to slightly improve over that. In contrast, datasets like Paracrawl and Toxicity are crawled and labeled with heuristics having different (and likely systematic) noise patterns than the “human noise”, and hence are easier tasks for any filtering and even more so for AutoCL. Overall, we believe our analysis and the conclusions we draw from it will make the present discussion around data noise and the role of filtering more fine-grained and as a result more insightful.
>
> Questions:
> 1. We agree, and in fact that was one of the motivations for AutoCL – throwing away a datum just because it looks like an outlier doesn’t sound right and the network should be given a chance to learn from it, in particular because it is likely to come from an underrepresented population. AutoCL detects, from our experiments, what data has most utility at each training step - and in a few cases, using the high self-influence examples provided the most utility, while low self-influence ones were likely learned very quickly.
>
> 2. While this is true, it is good to note that the sensitivity to model states across all layer settings is significantly smaller than to size/architecture (relative change in 90th $\cap$ is 12% vs. >30%). We haven’t explored the differences between the layers in detail, but we believe the situation you describe could be due to the “cancellation effects” ([Yeh et al., 2022](https://arxiv.org/abs/2202.11844)), where different examples share logic in the last layers and lead to a weaker discriminative power of the influence scores calculated via them. In contrast, the first/all layers cover low-level processing and word embeddings, permitting examples to reflect their own unique logic in the IF scores and be less susceptible to the cancellation effect. The bedrock changes in size or architecture could amplify the cancellation effect when first layers are not included and hence the overlap drops relatively by over 30% when comparing to first/all. The less foundational changes in model states, and in this case aided by the low prediction churn, happen to have an opposite effect but the difference is only ~12%.
>
> 3. We think the bulk of the difference in filtering performance could be explained by the ease of detecting some of synthetic noise – e.g. unrelated target or distantly wrong label – while natural noise is more subtle and ambiguous even for human annotators. For NQ, which is a well-curated dataset, due to such subtleties, the agreement between expert’s annotations of non-null answers and the consensus reached by the 5-way annotations were low: Figure 4 in the [NQ paper](https://aclanthology.org/Q19-1026) shows that debatable answers feature prominently in every annotator bin; according to Table 2 there, only <60% of answers are unambiguously correct after reannotation; finally, their Table 3 evaluates short-answer agreement precision/recall at only 63.4/52.6. In summary (and eyeballing the [released 5-way annotations](https://ai.google.com/research/NaturalQuestions/download)), many noisy examples are narrowly missing the correct answers (e.g. in terms of minimal correction to them) which requires common sense, cultural understanding and a certain degree of pedantism to spot.

---

### Official Review · Reviewer_Bjvq · 2023-08-12

**Soundness:** 3

**Excitement:**

3: Ambivalent: It has merits (e.g., it reports state-of-the-art results, the idea is nice), but there are key weaknesses (e.g., it describes incremental work), and it can significantly benefit from another round of revision. However, I won't object to accepting it if my co-reviewers champion it.

**Paper Topic And Main Contributions:**

This paper focuses on identifying noisy data/outliers and filtering out harmful instances to improve model performance. The author proposes using self-influence scores for data filtering and incorporating bandit curriculum learning to automate the filtering process instead of using a fixed filtering threshold. The stability and effectiveness of the self-influence scores are analyzed.

**Reasons To Accept:**

1. The idea of using a self-influence score to identify outliers or harmful samples is inspiring. If removing a sample deteriorates the loss value on itself, then this sample should be different enough from the rest of the training data. Such samples could be mislabeled, ambiguous, or difficult samples, or out-of-distribution samples.
2. Automated filtering using bandit curriculum learning seems effective and avoids manually tuning the hyperparameters of the previous fixed threshold.
3. Both the stability and efficacy of self-influence are analyzed on diverse datasets, tasks, and models, demonstrating the generality of the proposed method.

**Reasons To Reject:**

1. The claim that the proposed self-influence score is stable with respect to training and model hyperparameters across architecture variations is not actually supported by the evidence in Table 2 and Table 3. It can be observed that the self-influence score varies greatly for different model architectures and training hyperparameters.
2. The effectiveness of the proposed self-influence score is doubtful. When demonstrating the efficacy of the self-influence score in Table 4, nearly half of the results are missing and replaced with '-'. This may raise doubts that only favorable results are picked, especially for the out-of-distribution test. It would be appreciated if the author could provide those missing results and justify why they were missed in the original table.
3. There is no comparison with other data filtering techniques, such as AUM, Data Cartography, or PVI. How does the performance compare with those existing data filtering techniques?
4. I agree with the other two reviewers that the paper is hard to read. A better presentation style and more clarity would be appreciated. Specific suggestions were well given by the other two reviewers.

R[1] Pleiss, Geoff, Tianyi Zhang, Ethan Elenberg, and Kilian Q. Weinberger. "Identifying mislabeled data using the area under the margin ranking." *Advances in Neural Information Processing Systems* 33 (2020): 17044-17056.

R[2] Swayamdipta, Swabha, Roy Schwartz, Nicholas Lourie, Yizhong Wang, Hannaneh Hajishirzi, Noah A. Smith, and Yejin Choi. "Dataset Cartography: Mapping and Diagnosing Datasets with Training Dynamics." In Proceedings of the 2020 Conference on Empirical Methods in Natural Language Processing (EMNLP), pp. 9275-9293. 2020.

R[3] Ethayarajh, Kawin, Yejin Choi, and Swabha Swayamdipta. "Understanding Dataset Difficulty with $\mathcalV $-Usable Information." In International Conference on Machine Learning, pp. 5988-6008. PMLR, 2022.

**Reproducibility:**

3: Could reproduce the results with some difficulty. The settings of parameters are underspecified or subjectively determined; the training/evaluation data are not widely available.

**Reviewer Confidence:**

2: Willing to defend my evaluation, but it is fairly likely that I missed some details, didn't understand some central points, or can't be sure about the novelty of the work.

---

> ### Author Rebuttal · Authors · 2023-08-29
>
> Thank you for your comments.
>
> 1. Please note that w.r.t. the stability analysis, our contribution is that, to the best of our knowledge, we are the first to do such an analysis for self-influence scores, despite the IFs being used for filtering since 2017. As the results in Table 2 demonstrate, self-influence scores, unlike influence scores ([K & Sogaard, 2021](https://arxiv.org/abs/2111.04683)), are much more stable when the commonly modified *training* hyperparameters (batch size, data order, initialization) are used. However, when it comes to *model* parameters (Table 3), there is a substantial drop in stability. These are novel results and we believe they shed light on how to actually interpret highly self-influential examples: given these results, simply calling them “noise”, like done in prior work, is insufficient as they are apparently related to model type and capacity.
>
> 2. This is a serious misunderstanding: the missing values are either not possible to calculate for task/metric combinations in question or are not used in prior work (e.g. BLEU is not defined for toxicity classification etc.; likewise, F1 is not defined on Paracrawl and so on). For the given tasks and metrics, we provide all the results that apply and are required for comparison with prior work. We had to lump them into one table to save space and will split Table 4 into multiple tables to avoid the impression that the values were omitted.
>
> 3. Thank you for the pointers. Please note that we do cite two of the works mentioned – AUM (Pleiss et al. 2020) and dataset cartography (Swayamdipta et al., 2020) in the Related Work section as prior developments for dataset error detection that use training data dynamics. However, those methods are designed primarily for *classification* tasks, while in our work we mostly focus on seq2seq tasks (except for toxicity). Seq2seq tasks require beam search that would complicate calculation and require an adaptation of the mentioned methods. We thus opt for a task-agnostic method that can generalize across NLP tasks and investigate its properties as a prerequisite to use it effectively with AutoCL.
> Even more importantly, please note that we are not proposing a new *filtering* method – we fill the gaps in the analysis of an existing method and propose to go beyond filtering with AutoCL. That is, our main point should stand regardless of the filtering baseline: AutoCL should improve by using useful bits of information from the to-be-discarded examples and, in the least favorable case, AutoCL (given access to the buckets based on the same filtering scores) should agree with the threshold-based filtering.
>
> 4. Thank you, we will do our best to revise the style and content order.

---

### Meta-Review · Area_Chair_wVqc · 2023-09-19

**Recommendation:** 4

**Metareview:**

This paper studies the stability of self-influence scores, which determine memorizability and are task-agnostic, to training and model hyperparameters. Not only does self-influence have intrinsic properties that makes it desirable, the paper also aims to determine its capability towards cleaning noisy NLP datasets by removing outliers that have high self-influence.

The reviewers like the novelty of the idea and found it interesting, the application strongly motivated. They also commended that the AutoCL method is great.

There were concerns about the empirical evidence, raised by the most critical reviewers. I believe the authors’ rebuttal adequately addressed the concerns. There were some other questions about synthetic / natural noise which also seem addressed. The most critical reviewer also was convinced by the author rebuttal, which clarified some of the misunderstanding about the empirical evidence. Overall, this idea seems to be fairly unique and novel and exciting, as reflected in the reviews.

---

### Decision · Program_Chairs · 2023-10-07

**Decision:**

Accept-Main

**Comment:**

This paper studies the stability of self-influence scores, which determine memorizability and are task-agnostic, to training and model hyperparameters. Not only does self-influence have intrinsic properties that makes it desirable, the paper also aims to determine its capability towards cleaning noisy NLP datasets by removing outliers that have high self-influence.

The reviewers like the novelty of the idea and found it interesting, the application strongly motivated. They also commended that the AutoCL method is great.

There were concerns about the empirical evidence, raised by the most critical reviewers. I believe the authors’ rebuttal adequately addressed the concerns. There were some other questions about synthetic / natural noise which also seem addressed. The most critical reviewer also was convinced by the author rebuttal, which clarified some of the misunderstanding about the empirical evidence. Overall, this idea seems to be fairly unique and novel and exciting, as reflected in the reviews.